# Patient Satisfaction and Perceived Quality of Care with Telemedicine in a Pediatric Gastroenterology Clinic

**Michael Love [1], Anna K. Hunter [1,2], Gillian Lam [1,2], Linda V. Muir [1,2] and Henry C. Lin [1,2,*]**

1   Division of Gastroenterology, Doernbecher Children's Hospital, Portland, OR 97239, USA; michael.love@ucsf.edu (M.L.); huntean@ohsu.edu (A.K.H.); lamg@ohsu.edu (G.L.); muirl@ohsu.edu (L.V.M.)
2   Department of Pediatrics, Oregon Health & Science University, Portland, OR 97239, USA
*   Correspondence: linhe@ohsu.edu; Tel.: +1-503-494-1078

**Abstract:** Introduction: The coronavirus disease 2019 pandemic necessitated a shift to telemedicine for many clinics. This study aimed to better understand patient perception regarding telemedicine visits in a pediatric subspecialty clinic and to describe differences in management provided virtually versus in-person. Materials and Methods: This survey study and chart review was conducted at the Doernbecher Children's Hospital gastroenterology outreach clinics from May to June, 2020. The main hospital is located in Portland, Oregon, with the outreach clinics located in Salem, Eugene, and Medford, Oregon. Families were surveyed within 2 weeks of their visit, with a 6-month follow up survey. Results: There were 111 respondents to the initial survey (34% response rate). The majority of patients had initial positive impressions of telemedicine, with 75% feeling that the quality of telemedicine visits were as good as or better than in-person visits. At 6 months, there were 80 respondents (34% response rate), and this positive impression persisted with 72% of families reporting no negatives from their telemedicine experience. New patients seen via telemedicine were prescribed medications more frequently than those seen in-person (73% versus 45%, $p = 0.02$). Discussion: Patients and families felt the benefits of telemedicine visits outweighed the limitations both initially and at 6-month follow up. Telemedicine offers an effective alternative for pediatric subspecialty care especially for select conditions and follow up visits. However, the more frequent prescriptions could reflect the adaptation of clinical practice with the telemedicine platform, and further studies are needed.

**Keywords:** telehealth; access to care; outreach

## 1. Introduction

Telemedicine has been studied as an option to provide pediatric healthcare over the past several decades, especially to provide improved care and reduced cost to rural communities [1–8]. Surveys of families who used telemedicine prior to the the coronavirus disease 2019 (COVID-19) pandemic were often mixed, with one survey in a childhood obesity clinic reporting that families had overall positive impressions of telemedicine and a willingness to try again [9]. A different survey of patients seen by a pediatric rheumatology clinic showed that 95% of these families preferred in-person visits, even when families had long drives to the subspecialty clinic [10]. The varying reported benefits of telemedicine along with the limitations of the clinical exam and the perceived cost of implementation have contributed to limited patient access to telemedicine in the past. However, the COVID-19 pandemic has necessitated the rapid conversion of many pediatric general and subspecialty clinics to telemedicine [11–20].

The Oregon Health & Science University (OSHU) pediatric gastroenterology service provides care to a catchment area that spans the state of Oregon along with parts of southern Washington and northern California. Access to care, especially subspecialty care, is a challenge, as many patients have to drive hours to come to their appointment, requiring

significant time away from work, school, and other family responsibilities. To address these challenges, the Division had been piloting the use of telemedicine since 2019, but the onset of the COVID-19 pandemic accelerated the implementation of this platform and highlighted the need for the effective utilization of telemedicine. There are very few reports on the utility of or patient satisfaction with telemedicine in pediatric gastroenterology, with most of the reports that do exist focusing on specific conditions or the general option of telemedicine to improve access to care [5,21,22].

The objective of this study were as follows: (1) to assess patient and caregiver perceptions about utilizing the telemedicine platform for the outpatient clinic visit, and (2) to describe differences in clinical management provided via telemedicine as compared to in-person clinic.

## 2. Materials and Methods

A survey study was conducted at Doernbecher Children's Hospital's Pediatric Gastroenterology ambulatory outreach clinic from May to June 2020, soon after the pediatric gastroenterology clinic adopted the telemedicine platform due to the COVID-19 pandemic. The outreach clinics are located in the cites of Salem, Riverbend, and Medford, which are located between a 1 to 4 h drive away from Doernbecher Children's Hospital, which is located in Portland, Oregon. Clinics are typically held on weekly to monthly intervals depending on the outreach location. In these outreach locations, the Division continued to offer in-person clinics while adding telemedicine options for new and follow-up patients.

The study subjects were the caregivers of patients, given their role in caring for and being a patient advocate in pediatrics. Caregivers were called and administered an oral survey within 2 weeks of their clinic appointment, and they were approached again for a 6-month follow-up telephone survey. The initial survey included questions on caregiver perception of the telemedicine platform, with a focus on perceived benefits and potential barriers to use, along with questions on patient and caregiver satisfaction with their clinic visit. In addition, for subjects seen via the telemedicine platform, there were specific questions on user-experience and overall satisfaction with the telemedicine platform. The 6-month follow-up survey asked subjects to reflect on their initial clinic visit and assessed their satisfaction with their clinic visit, while also surveying caregiver perception on the telemedicine platform to see if perceptions about telemedicine had changed in the 6 months following their visit.

In addition to these surveys, a chart review was performed for all patients to assess patient demographics, primary diagnosis, diagnostic workup including lab work, radiology studies, procedures, prescriptions, and referrals ordered, along with any recommendations for inpatient admission. Clinic notes were also reviewed for the management plan recommended for each patient seen in the outreach clinic during this time period.

Descriptive statistics were used to summarize demographic data. The comparison of the diagnoses and management plans for new patients seen in clinic versus virtually was performed via Fisher exact testing.

## 3. Results

A total of 232 patients were seen in the outreach clinics during the months of May and June 2020 by four different providers, including three physicians with between 4 to >20 years of experience and one nurse practitioner with 2 years of experience in pediatric gastroenterology. There were 156 patients seen via telemedicine and 76 seen in-person. There were 84 new patient and 148 return patient clinic visits, with a patient age distribution as follows: <1 year old (4%), 1–10 years old (46%), and >10 years old (50%). Of the patients who lived within 15 miles of their identified primary clinic, 66% (74/112) were seen via telemedicine, compared to 68% (82/120) of patients living >15 miles away from the primary clinic being seen by telemedicine (Table 1).

**Table 1.** Characteristics of patients seen in pediatric gastroenterology outreach clinics.

| | Telemedicine Clinic | In-Person Clinic | |
|---|---|---|---|
| Total | **156** | **76** | |
| New or Returning patient | | | $p = 0.67$ [+] |
| New Patient | 55 (35%) | 29 (38%) | |
| Return Patient | 101 (65%) | 47 (62%) | |
| Primary Clinic Location * | | | |
| Medford, OR | 31 (20%) | 0 (0%) | |
| Portland, OR | 17 (11%) | 0 (0%) | |
| Riverbend, OR | 38 (24%) | 54 (71%) | |
| Salem, OR | 70 (45%) | 22 (29%) | |
| Distance from Primary Clinic (miles) | | | $p = 0.78$ [+] |
| <15 miles | 74 (47%) | 38 (50%) | |
| >15 miles | 82 (53%) | 38 (50%) | |
| Patient Age (years) | | | $p = 0.23$ [+] |
| Less than 1 | 9 (6%) | 1 (1%) | |
| Between 1–10 | 68 (44%) | 39 (51%) | |
| Greater than 10 | 79 (50%) | 36 (47%) | |
| Survey Respondants | | | $p = 0.64$ [+] |
| Initial Survey | 75 (68%) | 36 (32%) | |
| 6-Month Survey | 57 (71%) | 23 (29%) | |

* Based on patient address and closest proximity to outreach clinic or patient preference for established patients. Chi-square test was not performed as two clinic locations were not available for in-person visits during the study period. [+] *p*-value from Fischer exact test.

The caregivers of the 232 patients were approached for participation in the study with 111 families responding to the initial survey for a 47% response rate. Of the respondents to the initial survey, 68% were seen via telemedicine and 32% had an in-person clinic visit. Families were considered non-responders to the initial survey if they declined to participate or did not answer after two phone calls.

In the initial survey for patients seen via telemedicine, the majority of families found the telemedicine visits easy to use, with 89% planning to use telemedicine again and 95% recommending it to others. In particular, 75% reported that the quality of telemedicine visits was either the same as or better than in-person visits. The main caregiver-reported benefits of the telemedicine platform included the time saved from less driving time (96%) and the decreased cost (76%). By comparison, the main caregiver-reported limitation of telemedicine was the inability of the provider to perform a physical exam virtually (48%), with 33% reporting technical audiovisual difficulties. Despite reported audiovisual difficulties, 88% of families reported receiving the requisite assistance needed to use the telemedicine platform.

In the initial survey for families seen in-person in clinic, the primary reasons for the in-person visit included provider preference based on patient diagnosis (31%) and patient preference to obtain a physical exam (31%). Only 2% cited lack of access to technology as a reason for refusing a telemedicine visit.

At the 6-month follow up survey, 80 families responded for a 34% response rate, with 71% of respondents (57) seen via telemedicine and 29% seen in-person. Regardless of being seen in-person or via telemedicine, 91% reported that their experience with their child's clinical care was better than or as they expected, with 93% reporting that their child's symptoms were better than or the same compared to 6 months ago. Overall, the majority of caregivers (95%) would recommend telemedicine as a good option for clinical care even outside the setting of a pandemic.

In the 6-month follow-up survey of the patients seen via telemedicine, 37% of felt that an in-person visit would have been better for their child's clinical care. However, given the context of the pandemic, only 19% of caregivers would have preferred an in-person visit in hindsight. Caregivers reported that the main advantage of the telemedicine platform included not having to drive (67%), with the ability to socially distance and more efficient service being other reported benefits (Table 2). The top disadvantages of the telemedicine

platform were the lack of a physical exam (56%), the lack of personal connection (26%), and technological glitches (14%). Of the patients seen by telemedicine, those who lived more than 15 miles from their primary clinic were more likely to report the positive impact of telemedicine on their child's care (77% vs. 54% for patients who lived within 15 miles from the clinic). Similarly, established patients were more likely to report the positive impact of telemedicine (77% vs. 59% for new patients).

**Table 2.** Pros and cons of virtual visits as noted by caregivers in the 6-month follow-up survey.

| Pros | Telemedicine: *n* (%) | In-Person: *n* (%) |
| --- | --- | --- |
| Lack of Travel | 38 (67%) | 12 (52%) |
| Efficiency | 11 (19%) | 6 (26%) |
| Ability to Socially Distance | 11 (19%) | 4 (17%) |
| Ease of Scheduling | 7 (12%) | 4 (17%) |
| Do Not Need Childcare | 7 (12%) | 3 (13%) |
| Patient Comfort | 3 (5%) | 2 (9%) |
| Do Not Miss School | 3 (5%) | 1 (4%) |
| Do Not Need Mask | 2 (4%) | 0 (0%) |
| Easier Follow Up | 1 (2%) | 0 (0%) |
| **Cons** | **Telemedicine:** *n*(%) | **In-Person:** *n*(%) |
| Lack of Physical Exam | 32 (56%) | 13 (57%) |
| Lack of Personal Connection | 15 (26%) | 5 (22%) |
| Technology Glitches | 8 (14%) | 3 (13%) |
| Poor Communication | 0 (0%) | 2 (9%) |
| Efficiency | 0 (0%) | 1 (4%) |
| Reluctant to Call Back | 1 (2%) | 0 (0%) |
| Inability to Include Multiple People | 1 (2%) | 0 (0%) |
| Finding Location for Telemedicine Visit | 1 (2%) | 0 (0%) |
| Patient Less Motivated | 1 (2%) | 0 (0%) |

In the 6-month follow up survey for patients seen in-person, all respondents felt that telemedicine should be an option for families even outside of a pandemic, with 91% having used the telemedicine platform at some time in the previous 6 months. Based on these experiences, 52% felt that telemedicine could provide the same level of care as an in-person clinic visit. The top three benefits and disadvantages of the telemedicine platform were similar to those identified by caregivers whose children was seen by telemedicine, with the primary advantage being not having to drive and the main disadvantage being the lack of a physical exam (Table 2). For patients who were seen in-person, 63% of new patients and 47% of established patients felt telemedicine would have likely provided the same level of care as an in-person visit.

In a retrospective chart review, the top clinical diagnoses for new patients seen both via the telemedicine platform or in-person were abdominal pain, constipation/encopresis, and diarrhea (Table 3). A review of the management based on labs, radiographic studies, procedures ordered, referrals placed, and inpatient admission revealed similar rates of each of these management modalities for patients seen in-person versus via telemedicine (Figure 1). This data was adjusted for multiple testing. There was no difference in the management for a patient seen via telemedicine or in-person by primary clinical diagnosis. When comparing the management for new patients, there was a statistically significant difference in the frequency of ordering prescriptions, with more patients seen via telemedicine having a prescription ordered for them (73% vs. 45%, *p*-value < 0.02).

**Table 3.** Primary diagnosis of new patients seen in pediatric gastroenterology outreach clinics.

| Diagnosis | Telemedicine | | In-Person | |
|---|---|---|---|---|
| | **New** | **Return** | **New** | **Return** |
| Abdominal Pain | 16 (29%) | 11 (11%) | 8 (28%) | 8 (17%) |
| Constipation | 12 (22%) | 25 (25%) | 2 (7%) | 3 (6%) |
| Diarrhea | 9 (16%) | 2 (2%) | 5 (17%) | 0 (0%) |
| Gastroesophageal Reflux | 6 (11%) | 18 (18%) | 1 (3%) | 2 (4%) |
| Nausea/Vomiting | 6 (11%) | 4 (4%) | 6 (21%) | 4 (9%) |
| Poor Weight Gain | 2 (4%) | 7 (7%) | 1 (3%) | 4 (9%) |
| Dysphagia | 2 (4%) | 7 (7%) | 2 (7%) | 2 (4%) |
| Inflammatory Bowel Disease | 0 (0%) | 10 (10%) | 0 (0%) | 12 (26%) |
| Celiac Disease/EoE/Allergy | 1 (2%) | 7 (7%) | 1 (3%) | 4 (9%) |
| Elevated Liver Enzymes | 0 (0%) | 3 (3%) | 1 (3%) | 1 (2%) |
| Liver Disase/Liver Transplant | 0 (0%) | | 1 (3%) | 2 (4%) |
| Other * | 1 (2%) | 7 (7%) | 1 (3%) | 5 (11%) |

\* Other diagnoses included: abnormal imaging, genetic condition with GI involvement, iron deficiency anemia, H pylori, short gut syndrome, fructose intolerance, pancreatitis, abnormal imaging, and pelvic floor dysfunction.

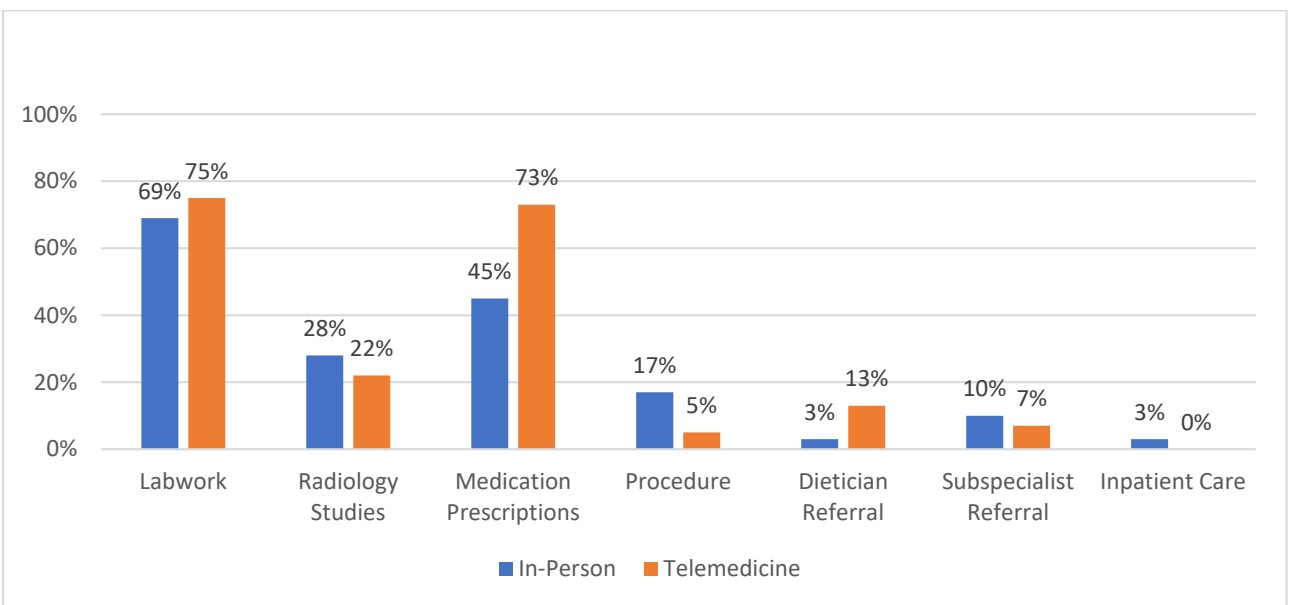

**Figure 1.** Comparison of work-up and management between patients seen in-person and via telemedicine.

## 4. Discussion

This study on pediatric caregiver perceptions on the telemedicine platform demonstrates a general openness to the use of telemedicine in the pediatric gastroenterology setting. The study population consisted of patients who were scheduled to be seen at outreach clinic sites for whom access to subspecialty care may be a challenge. While the outreach clinics have traditionally been in-person clinics and are located in areas to make access to academic center subspecialty care easier, some patients may still have long commutes to the outreach clinics. Telemedicine offers a practical solution for patients for whom transportation and the associated costs is a significant barrier or inconvenience to receiving subspecialty care. Our pediatric gastroenterology team has been using the telemedicine platform to better provide outreach clinical care, as over 25% of patients live >40 miles from a pediatric gastroenterology clinic [23]. During the beginning of the COVID-19 pandemic, OHSU pediatric gastroenterology was able to quickly implement an effective telemedicine clinic to help patients socially distance. One of the main challenges with telemedicine is to ensure appropriate technical requirements for both the clinic and patient. Additionally, patient education for access and utilizing the telemedicine platform can help facilitate ease of

use [24,25], and at our institution, a team of nurses, schedulers, and information technology specialists were integral in helping patients learn to navigate the telemedicine platform. In our study, the majority of families reported that the telemedicine platform was easy to use, and families who experienced technical difficulties received the requisite assistance.

Most families had a positive impression of telemedicine regardless of whether they were seen in-person of via telemedicine, suggesting that families feel that there is a role for telemedicine to provide care for select pediatric gastroenterology patients. It is possible that this perception was influenced by external factors such as personal safety and social distancing due to the COVID-19 pandemic. The utilization of the telemedicine platform has grown dramatically since the pandemic, with many studies reporting patient satisfaction with the telemedicine platform [26–30]. Similarly, in our study, most families who used the telemedicine platform were satisfied with their experience and would recommend it to others. In addition, almost all the families surveyed felt that telemedicine should be an option for families even outside of the COVID-19 pandemic. When discussing the benefits of telemedicine, most patients focused on the time and cost savings relating to transportation to and from the appointment, suggesting that these factors play a role in access to subspecialty care for our patient population. Having a modality to allow for the establishment of quality care such as telemedicine can help to address some of these access challenges. While there may be concerns with the ability to establish a relationship and meaningfully address the patient's issue via telemedicine, studies have reported that patients feel that telemedicine can be effective in establishing care [31] and providing gastroenterology specific care [32,33], although there are limited studies on using the telemedicine platform to establish a pediatric gastroenterology diagnosis. In our study, only 4% of patients seen by telemedicine required hospitalization for further management of GI-related symptoms in the following 6 months, with some of these admissions being directed based on the initial telemedicine consultation. As such, it is possible that telemedicine offers a sufficient platform for pediatric gastroenterology care in many conditions.

Despite the overall positive impression towards telemedicine, on the 6-month follow up survey, 37% of those seen via telemedicine felt that an in-person visit would have been a preferable clinic modality in hindsight. The most consistent reasons for a preference towards in-person visits seems to have been the desire to obtain a physical exam, as well as the perceived lack of personal connection with the provider in a telemedicine visit. In our study, strategies employed by providers to examine the patient via the telemedicine platform included visual inspection and having the patient or caregiver palpate the abdomen to elicit tenderness. However, efforts to examine patients via video are difficult when patients are experiencing symptoms such as abdominal pain, and parents may have felt that serious diagnoses could not be ruled out without an in-person physical exam. Given these physical exam limitations, most of the information gathered for medical decision making likely came from the patient history, thus placing additional emphasis on communication. Some families also identified a lack of in-person communication by a physician as a hesitation to use the telemedicine platform. This concern highlights the need to continue to educate and assess how the provider can establish a relationship and communicate effectively with telemedicine. It is also imperative to assess if select pediatric gastroenterology diagnoses might be more amenable to be seen via telemedicine either as an initial visit or for follow-up.

Other caregiver preferences with telemedicine that were noted in our study include the observation that caregivers of new patients seen via telemedicine were less likely to identify the lack of a physical exam as a potential barrier, which suggests that the physical exam may be perceived as more important by families who had previously had an in-person clinic visit. In addition, patients who lived further than 15 miles from the clinic were more likely to favor telemedicine, suggesting that the benefits of decreased travel time and associated cost saving might eclipse any perceived negatives of telemedicine. With this perceived convenience provided by the telemedicine platform, it is important to also maintain a similar level of care to in-person visits.

In our study, the management of patients for select gastrointestinal conditions was similar regardless of whether they were seen in-person or via telemedicine. The similar practice pattern of ordering diagnostic tests suggests that providers feel comfortable enough with the history obtained to initiate management despite the physical exam limitations via telemedicine. The similar frequency and types of labs and procedures ordered by diagnosis suggests that part of the initial clinical management could be primarily driven by the patient history. It is possible that physical exam findings can inform different management decisions than otherwise indicated by labs and procedures. The prescription of medications was one area that differed between in-person and telemedicine. It is possible that the lack of a physical exam made it harder for providers to determine the initial management, thus leading to more frequent empiric trials of medications to help manage symptoms. An increase in prescribing medications to patients seen via telemedicine when compared to previous in-person practice patterns has been noted in specialties such as in interventional pain management [34], but other specialties, such as sports medicine, have reported a decrease in prescriptions [35]. This variable practice pattern suggests that such prescribing practices could be a reflection of patient and provider preferences or limited in-person access in response to COVID.

The limitations of this study include response bias from this survey study. In addition, as this was a sample of convenience, the relatively small sample size, lack of power, and lack of randomized, matched study groups made it difficult to make ideal comparisons or generalize the results for any single diagnosis. Another limitation is population bias, as this study population was of families seen in the outreach clinics who were more likely to have less access, longer travel times to the clinic, and perhaps a predilection towards telemedicine.

## 5. Conclusions

In conclusion, telemedicine offers an effective alternative for pediatric gastroenterology care. Patients and their families generally reported that the benefits of telemedicine outweighed the limitations, with key advantages being improved access and decreased costs while maintaining a similar initial quality of care. With the main limitation of the telemedicine platform being the inability to perform a comprehensive physical exam, it can be helpful to have clear guidelines or distinctions for which type of diagnoses or patients can be best served by the telemedicine platform. It is possible that telemedicine is well suited for subspecialty follow-up care or the integration of multi-disciplinary care in which it can be difficult to arrange the schedule of different providers for an in-person clinic. Ultimately, effective communication with the patient and family is key in helping to overcome the potential barriers posed by the telemedicine platform. To improve the utilization of telemedicine by providers in medical decision making, dedicated training on communication considerations unique to telemedicine may be helpful.

Further studies are needed to optimize care provided via the telemedicine platform and to objectively assess the impact or differences in the management and outcomes of specific diagnoses. In particular, our future directions include understanding the drivers of any differences in management, such as the recommendation of diagnostic testing or the prescription of medications. With a larger study cohort, a cost-benefit analysis can be also considered. As telemedicine becomes more integrated into routine medical practice, it is imperative that we continue to understand how best to utilize this platform.

**Author Contributions:** M.L. contributed to the data collection, analysis, and manuscript writing and revision. A.K.H. contributed to the study conception, data collection, and manuscript revision. G.L. contributed to the study conception, data collection, and manuscript revision. L.V.M. contributed to the study conception, data analysis, and manuscript writing and revision. H.C.L. contributed to the study conception, data analysis, and manuscript writing and revision. All authors have read and agreed to the published version of the manuscript.

**Funding:** There was no funding support for this study.

**Institutional Review Board Statement:** This study was approved by the Oregon Health & Science University Institutional Review Board (Study00021587).

**Informed Consent Statement:** Informed consent was obtained from all subjects involved in the study.

**Data Availability Statement:** The authors confirm that the data supporting the findings of this study are available within the article.

**Conflicts of Interest:** The authors declare no conflict of interest.

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
