# Peer review of "Patient Satisfaction and Perceived Quality of Care with Telemedicine in a Pediatric Gastroenterology Clinic"

_pediatrrep, doi:10.3390/pediatric14020025_

Round 1
Reviewer 1 Report
This paper touched on the results of a survey of caregivers of children with gastroenterological diseases who underwent a telemedicine examination for two months during the COVID 2019 pandemic. A distinctive feature of the survey is that it was conducted in two stages with a difference of six months, which allows for more accurate determine the opinion and mood of educators regarding telemedicine services. The paper does not contain grammatical and linguistic flaws, the quality of writing and presentation of thoughts is satisfactory.
• It is required to rewrite the abstract of the paper. The abstract should give generalized information about the paper. However, in this paper, we noticed that all the main results of the paper are given in the abstract, after reading it, the reader receives full information about the results, and there is an impression that the rest of the paper simply repeats the information given in the abstract. Also, try to write the abstract in one whole chapter without breaking into parts like Introduction, Materials and Methods, Results and Discussion.
• The paper deals with telemedicine services, but there is no specific information on the types of services. Questions remain: What kind of telemedicine services are we talking about?, How was the telemedicine examination carried out?, What diagnostic methods or examinations were carried out?, What equipment was used in the examination on the patient side and on the service provider side. Try to devote one chapter or subchapter to these questions.
• To make the paper more useful from a scientific point of view, the age distribution of the examined children should be given in one of the chapters. Include another table with the age of the examined children, the types of diseases detected during the telemedicine examination and the satisfaction of the patients themselves (if possible).
• If possible, include in the paper information about the medical staff who conducted telemedicine examinations, since age, gender, experience, and other personal qualities also directly affect the level of patient satisfaction.
Author Response
It is required to rewrite the abstract of the paper. The abstract should give generalized information about the paper. However, in this paper, we noticed that all the main results of the paper are given in the abstract, after reading it, the reader receives full information about the results, and there is an impression that the rest of the paper simply repeats the information given in the abstract. Also, try to write the abstract in one whole chapter without breaking into parts like Introduction, Materials and Methods, Results and Discussion.
- Thank you for this feedback. The abstract was written with distinctive sections to help the reader follow the abstract. As for rewriting the abstract, our understanding is that an abstract should highlight the key points of the study which include a general positive perception of families about the use of telemedicine. However, we did not include information about the other main observation that prescription medication were ordered more frequently in patients seen via telemedicine.
The paper deals with telemedicine services, but there is no specific information on the types of services. Questions remain: What kind of telemedicine services are we talking about?, How was the telemedicine examination carried out?, What diagnostic methods or examinations were carried out?, What equipment was used in the examination on the patient side and on the service provider side. Try to devote one chapter or subchapter to these questions.
- The telemedicine services include the traditional video set up to allow for real-time video interaction between physician and patient. The patient could download the app for this platform onto their phone or computer, as long as they had video capability. Conducting a comprehensive physical exam, or even a portion of a physical exam has historically been a limitation of telemedicine which presumptively had been one of the barriers of adoption. For pediatric gastroenterology, examination techniques include a visual inspection by the provider and having the patient or caregiver palpate the abdomen. We have included this information in the discussion session. [Page 7, Paragraph 1]
To make the paper more useful from a scientific point of view, the age distribution of the examined children should be given in one of the chapters. Include another table with the age of the examined children, the types of diseases detected during the telemedicine examination and the satisfaction of the patients themselves (if possible).
- We have included the age distribution of the patient in Table 1 and in the results section. [Page 3, Paragraph 1]
If possible, include in the paper information about the medical staff who conducted telemedicine examinations, since age, gender, experience, and other personal qualities also directly affect the level of patient satisfaction.
- We have included provider demographics into the results section. [Page 3, Paragraph 1]
Reviewer 2 Report
The paper presents a scientific study about the comparison between the quality of care of a telemedicine solution and the traditional visit for pediatric gastroenterology.
The paper is well structured, but many details should be added.
Starting from the aims initially defined by the authors:
1) "to assess patient and caregiver perception about utilizing the telemedicine platform for the outpatient clinic visit"
What does it mean "perception" in this case? Are you speacking about user-experience, satisfaction, usability? According to which parametera the authors want to assess, there are specific questionnaries that should be used. The paper only speack about "surveys", but there is no descrption about the specific information evaluated by the quetionnaries, both for patients and clinicians. A deteilad description about surveys should be addedd.
Furthermore, how are the key figures behind the research study? When a new telemedicine platform is used, there are also nurses and IT personnel who play a crucial role to improve the satisfaction of the end-user.
Some information about the specific telemedicine should be added.
2) "to assess for differences in clinical management provided via telemedicine as compared to in-person clinic."
This aim is not so reached by the results presented. Also, reaching this results requires another scientific paper spefic for this aim. For example, a cost benefit analysis should be reported. Also, a worflow analysis shouold be mapped to compare the traditional approach with the telemedicine.
I wuold like to suggest a review of the second aim.
The conoclusion are not enough. Much more information should be discussed. A good solution should be to embed the conclusion in the final discussion.
Author Response
Starting from the aims initially defined by the authors:
1) "to assess patient and caregiver perception about utilizing the telemedicine platform for the outpatient clinic visit"
What does it mean "perception" in this case? Are you speaking about user-experience, satisfaction, usability? According to which parameters the authors want to assess, there are specific questionnaires that should be used. The paper only speak about "surveys", but there is no description about the specific information evaluated by the questionnaires, both for patients and clinicians. A detailed description about surveys should be added.
- Thank you for this feedback. The first aim of the study is to assess if there are any existing perceptions about using telemedicine. These perceptions can include the perceived usability and any potential barriers or limitations to telemedicine compared to a traditional in-person visit. As the study population included a people who were seen either via the telemedicine platform or in-person, the surveys were also designed to assess their satisfaction and user-experience with the clinical visit. We have included additional information into the “Materials and Methods” section to help clarify. [Page 2, Paragraph 3]
Furthermore, how are the key figures behind the research study? When a new telemedicine platform is used, there are also nurses and IT personnel who play a crucial role to improve the satisfaction of the end-user. Some information about the specific telemedicine should be added.
- To establish a telemedicine platform, IT personnel, schedulers, medical assistant, and nurses played crucial roles in ensuring the usability of the platform. We have included some of this information in the discussion. [Page 6, Paragraph 1]
2) "to assess for differences in clinical management provided via telemedicine as compared to in-person clinic."
This aim is not so reached by the results presented. Also, reaching this results requires another scientific paper specific for this aim. For example, a cost benefit analysis should be reported. Also, a workflow analysis should be mapped to compare the traditional approach with the telemedicine. I would like to suggest a review of the second aim.
- Thank you for the feedback. A larger and more in-depth study is needed to fully assess any differences in management via telemedicine versus in-person platform. Our goal is to initially describe any differences to set the stage for future study. We have updated the aims to reflect this intention [Page 2, Paragraph 1] as well as include information about future directions in the conclusion. [Page 2, Paragraph 1]
The conclusion are not enough. Much more information should be discussed. A good solution should be to embed the conclusion in the final discussion.
- We have included additional information in the conclusion. However, we have held off on making a final recommendation as this is a pilot study of patient perceptions of the telemedicine platform and as such, is not meant to come up with a solution. [Page 8, Paragraph 2]
Reviewer 3 Report
Report on “Patient satisfaction and perceived quality of care with telemed-icine in a pediatric gastroenterology clinic”
Manuscript Number: pediatrrep-1637248
Pediatric Reports
Summary
The article evaluates the patients and caregivers’ perception about using a telemedicine platform for the outpatient clinic visit. Further, it examines differences in clinical management provided via telemedicine as compared to in-person clinic.
Assessment
The article’s aim addresses an important research gap, i.e. the acceptance and effectiveness of tele-medical outpatient care. While there have been several studies comparing care provided via telemedicine and in-person, the assessment of the patient’s perspective is underrepresented in literature. Therefore, the findings of this study enrich the empirical evidence.
The used data set offers a valid approach to address the research questions. However, the presentation of the results lack of clarity making it hard for the reader to follow.
In what follows, I give a couple of suggestions that may improve the paper.
Abstract
- Materials and Methods: Please provide the location of the hospitals, e.g. the city or the state.
- Results: Please add the number of respondents and the response rate for both surveys (2 week and 6-month follow up)
- Results: “…75% feeling the visits were as…” What visits do you mean, telemedicine or in-person? Either way, the meaning of the sentence is not clear to me. Please revise.
Introduction
- The introduction is well written and comes precisely to the point.
Materials and Methods
- It is not clear, whether you also asked the caregivers that treated the patients or whether the caregivers’ children were treated and they were asked because of their role as parents. The statement on page 4 in the first paragraph (“by caregivers whose child”) makes me think that the latter might be true. Please clarify this ambiguity by explaining in the methods section in more detail who the caregivers are, their role, and how and when they were surveyed. If the first assumption (caregivers have treated the patients) is true, then please also state, why were they only asked after 6 month and not also after 2 weeks. What is the motivation for this?
- Why did you perform Fisher exact testing only for the diagnosis and management plan? Whether there are significant differences between telemedicine and in-person care would also be interesting for the other variables for that you show summary statistics in Table 1 and 2.
- Did you adjust for multiple testing in Figure 1?
Results
- How many caregivers were surveyed? Please also add the response rate of this group.
- Table 1: I would suggest adding the Fisher’s exact p-value for differences among the subgroups in an extra column.
- Page 3: Please also show the numbers of the initial survey for patients seen via telemedicine as well as the 6-month follow-up survey in a table. This makes it easier for the reader to follow. I guess all the numbers you provide on page 3 would be able to provide in a single table by comparing, e.g., the initial and the 6-month follow-up survey.
- Table 3: Why are there so small numbers of observations? Is it because most cases were treated without assigning a diagnosis? If so, I would expect that the number of cases with an assigned diagnosis would be higher for the in-person care in comparison with the telemedicine care. It would be helpful to provide this number in Table 1.
Discussion
- The finding that the prescription of medications were differed between in-person and telemedicine is very interesting. Please add how this finding is related to literature.
- Please add the lack of power due to the small sample size as a further limitation of your study. This makes it hard to identify further differences between in-person and telemedicine.
Author Response
Abstract
- Materials and Methods: Please provide the location of the hospitals, e.g. the city or the state.
- The study was performed at the hospital and clinics located in Salem, OR, Eugene, OR, and Medford, OR. This has been added to the abstract. [Page 1, Paragraph 1]
- Results: Please add the number of respondents and the response rate for both surveys (2 week and 6-month follow up)
- The initial response rate was 47% and the follow-up survey had a 34% response rate. This has been added to the abstract. [Page 1, Paragraph 1]
- Results: “…75% feeling the visits were as…” What visits do you mean, telemedicine or in-person? Either way, the meaning of the sentence is not clear to me. Please revise.
- The sentence should read that 75% of respondent felt that telemedicine visits were as good or better than in-person visits. This has been revised in the abstract. [Page 1, Paragraph 1]
Introduction
- The introduction is well written and comes precisely to the point.
Materials and Methods
- It is not clear, whether you also asked the caregivers that treated the patients or whether the caregivers’ children were treated and they were asked because of their role as parents. The statement on page 4 in the first paragraph (“by caregivers whose child”) makes me think that the latter might be true. Please clarify this ambiguity by explaining in the methods section in more detail who the caregivers are, their role, and how and when they were surveyed. If the first assumption (caregivers have treated the patients) is true, then please also state, why were they only asked after 6 month and not also after 2 weeks. What is the motivation for this?
- Study subjects were caregivers of patients given their role in caring for and being a patient advocate in pediatrics. This has been added to the Materials section. [Page 2, Paragraph 2]
- Why did you perform Fisher exact testing only for the diagnosis and management plan? Whether there are significant differences between telemedicine and in-person care would also be interesting for the other variables for that you show summary statistics in Table 1 and 2.
- We did perform the other analyses as you mentioned and have included that information in Table 1. We did not perform a Fischer exact test on primary clinic location as during the study period, 2 clinic sites were on modified operations which impacted the scheduling of in-person clinics.
- Did you adjust for multiple testing in Figure 1?
- The data was adjusted for multiple testing. This has been added to the results section. [Page 5, Paragraph 0]
Results
- How many caregivers were surveyed? Please also add the response rate of this group.
- A total of 232 caregivers were approached for the study with 111 responding to the initial survey (47%) response rate and 80 responded to the 6-month follow-up survey for a 34% response rate. This information is in the manuscript and has been clarified. [Page 3, Paragraph 1 & 4]
- Table 1: I would suggest adding the Fisher’s exact p-value for differences among the subgroups in an extra column.
- We have included this data in Table 1 as suggested. [Page 3]
- Page 3: Please also show the numbers of the initial survey for patients seen via telemedicine as well as the 6-month follow-up survey in a table. This makes it easier for the reader to follow. I guess all the numbers you provide on page 3 would be able to provide in a single table by comparing, e.g., the initial and the 6-month follow-up survey.
- Thank you for this feedback about presenting the data more clearly. We have incorporated this feedback into the Table. [Page 3]
- Table 3: Why are there so small numbers of observations? Is it because most cases were treated without assigning a diagnosis? If so, I would expect that the number of cases with an assigned diagnosis would be higher for the in-person care in comparison with the telemedicine care. It would be helpful to provide this number in Table 1.
- Table 3 was intended to show the diagnoses for new patients seen in the clinic. The goal was to identify what type of diagnoses providers and caregivers felt were amenable to be seen via telemedicine. We have revised the Table and also included diagnoses of return patients. [Page 5]
Discussion
- The finding that the prescription of medications were differed between in-person and telemedicine is very interesting. Please add how this finding is related to literature.
- There is variable practice when it comes to prescribing medications and we have included this information in the discussion. [Page 7, Paragraph 3]
- Please add the lack of power due to the small sample size as a further limitation of your study. This makes it hard to identify further differences between in-person and telemedicine.
- We agree that the lack of power due to the small sample size is a limitation of the study and this has been added to the manuscript. [Page 7, Paragraph 3]